# Molecular Detection of Rabies Lyssaviruses from Dogs in Southeastern Nigeria: Evidence of TransboundaryTransmission of Rabies in West Africa

**DOI:** 10.3390/v12020134

**Published:** 2020-01-23

**Authors:** Ukamaka U Eze, Ernest C Ngoepe, Boniface M Anene, Romanus C Ezeokonkwo, Chika I Nwosuh, Claude T Sabeta

**Affiliations:** 1Department of Veterinary Medicine, Faculty of Veterinary Medicine, University of Nigeria, Nsukka 410001, Nigeria; boniface.anene@unn.edu.ng; 2Agricultural Research Council-Onderstepoort Veterinary Institute, OIE Rabies Reference Laboratory, Onderstepoort 0110, South Africa; ngoepee@arc.agric.za (E.C.N.); SabetaC@arc.agric.za (C.T.S.); 3Department of Veterinary Parasitology and Entomology, Faculty of Veterinary Medicine, University of Nigeria, Nsukka 41001, Nigeria; romanus.ezeokonkwo@unn.edu.ng; 4National Veterinary Research Institute, Vom, Plateau State 930103, Nigeria; chikanwosuh@gmail.com; 5Department of Veterinary Tropical Diseases, University of Pretoria, Onderstepoort 0110, South Africa

**Keywords:** rabies lyssavirus, dogs, molecular characterization, southeastern Nigeria, transboundary transmission

## Abstract

Despite being the first country to register confirmed cases of Mokola and Lagos bat lyssaviruses (two very distant lyssaviruses), knowledge gaps, particularly on the molecular epidemiology of lyssaviruses, still exist in Nigeria. A total of 278 specimens were collected from dogs in southeastern Nigeria between October 2015 and July 2016, and 23 (8.3%) of these tested positive for lyssaviruses with the direct fluorescent antibody test (DFA). The lyssaviruses were genetically characterized by amplifying the highly conserved nucleoprotein (N) gene of the rabies lyssaviruses (RABVs) of the viral genome. Phylogenetic analyses of the nucleotide sequences showed that all the RABV sequences in this study were of the Africa-2 lineage. Our results demonstrated that transboundary transmission of rabies lyssavirus is a key event, given that one of the RABV sequences (MN196576) clustered with rabies variants from neighboring Niger Republic. Furthermore, three RABVs from dogs from Anambra State clustered separately forming a novel and distinct group. Our results demonstrated that transboundary transmission of RABLVs is a key driver in the spread of rabies in West Africa. In order for the successful control of this zoonotic disease, a multinational stepwise surveillance and elimination of rabies in Africa by 2030 is probably the solution for regional elimination.

## 1. Introduction

Rabies is one of the oldest but deadliest zoonotic diseases known to man, and has a case fatality rate approaching 100% once clinical symptoms are apparent. The first report of the disease in Nigeria dates back to about 100 years ago, and since then rabies has been endemic in this West African country, resulting in approximately 1637 annual human deaths [1,2]. The causative agent of rabies is a highly neurotropic virus and member of the 16 currently recognized viral species of the *Lyssavirus* genus [3,4]. Apart from RABV, which is mainly dog-mediated through bites but also scratches, there are other lyssaviruses that were identified in the country in the late 1950s, namely Lagos bat lyssavirus (LBLV) and Mokolalyssavirus (MOKL), making this country an important place for lyssavirus origin and evolution [5,6]. These two, LBLV and MOKL, have not been identified since the late 1950s, underlining the lack of surveillance activities for these viruses.

Nigeria is a large country and a member of the 16 West African regional network, a region that covers a total land area of about 5,112,903 km^2^ [7], with an estimated population of about 381,981,000 as of 2017 [8]. The boundaries within the West African countries are porous (as in most parts of the African continent) and allow all sorts of activities such as smuggling, human, drug, and illicit arms trafficking, nomadism, and dog trading, which may promote rabies virus transmission [9,10]. The porosity of the border areas facilitates transboundary transmission of animal diseases such as rabies as vectors move easily between neighboring countries. The illegal trade of dogs within and across West African countries is a contributory factor to the transboundary movement of the disease, a phenomenonnot only unique to Africa [11,12,13,14,15,16,17]. In the sub-Saharan African region, three genetically distinct rabies lyssavirus lineages (Africa 1, 2, and 3) were identified, each spanning a different region [18]. Initially, African sub lineage 1a and lineage 2 occurred mainly in the West African countries [19]. However, further analyses showed that the rabies lyssavirus belonging to African lineage 1b, initially thought to be exclusively found in eastern, central, and southern Africa, now occurs in West Africa [13], highlighting the complexity of rabies epidemiology in the region. Furthermore, a rabies lyssavirus strain in Liberia was found to cluster with other rabies viruses in the China lineage 2, confirming transcontinental transmission of the disease [20].

The reverse-transcription polymerase chain reaction (RT-PCR), including the sequencing of specific gene products and reconstruction of phylogenetic trees, are now very important tools used in monitoring disease epidemiology and surveillance [21]. The rabies lyssaviruses from northern and western parts of Nigeria have been previously characterized [12,22,23,24], but only four rabies lyssaviruses from dogs were characterized from Enugu (Southeastern Nigerian) [25]. The scant molecular data available was ample motivation to genetically characterize additional rabies lyssaviruses from Southeastern Nigeria. Furthermore, rabies is a highly under-reported disease in the region, as many rabies cases are not presented to veterinary clinics and hospitals, and we therefore decided to survey the dog markets for possible rabies-infected dogs and assess the public health burden of the disease. The study was therefore designed to detect RABVs in dogs and determine the phylogenetic relationships of the rabies lyssaviruses from Southeastern Nigeria, neighboring countries (Cameroon, Chad, Niger, and Benin), and other regions by comparing the nucleoprotein gene nucleotide sequences.

## 2. Materials and Methods

### 2.1. Ethical Approval

The ethical approval (UNN/eTC/14/68625) for the molecular investigation of rabies lyssaviruses in southeastern Nigeria was granted by the University of Nigeria Ethical Committee on the 17th of August, 2015. The brain specimens were collected with the consent of dog traders, dog meat sellers, and pet owners.

### 2.2. Study Locations

The study was conducted in the Southeastern geopolitical zone of Nigeria, formerly known as the eastern region (Figure 1). Southeastern Nigeria, also known as Igboland (because the common language is Igbo language), is made up of five States, covering a total land area of 17,545 km^2^ with an estimated population of 40 million [26]. The region is located between latitudes 7°07′ N and 3°90′ N, and longitudes 6°51′ E and 8°30′ E. This area is characterized by a tropical climate with a distinct wet season that lasts from April to October, and a dry season that lasts from November to March, each year. Three (Anambra, Ebonyi, and Enugu) of the five States of the geopolitical zones were selected by random sampling and included in the study. The dog markets and restaurants in each of the States which are located in the rural areas were selected purposively based on availability of slaughtered dogs for human consumption, while the clinic location in urban area was based on presentation of suspected rabies cases. Given that there is currently no rabies diagnostic laboratory in Southeastern Nigeria, all the rabies specimens were sent to National Veterinary Research Institute (NVRI), Vom, Plateau State, which is the national rabies reference laboratory.

### 2.3. Specimen Collection

The specimens were collected over 9 months, between October 2015 and July 2016. During this period, a total of 278 dog heads were collected from dog markets, restaurants, and veterinary clinics. All the dog heads were transported to the postmortem room, Department of Veterinary Pathology and Microbiology, University of Nigeria, Nsukka. The composite brain parts (comprising the hippocampus, brain stem, cerebellum, and cerebrum) were extracted and stored in leak-proof containers. All the samples were stored frozen at (−20 °C) until analyzed.

### 2.4. Direct Fluorescent Antibody Test for Detection of Rabies Lyssavirus Antigen

The brain specimens were initially transported to the National Veterinary Research Institute, Vom (NVRI) at 4 °C for initial rabies diagnosis using the direct fluorescent antibody test (DFA) [27] using a fluorescein isothiocyanate (FITC) anti-rabies nucleocapsid conjugate (Fujirebio^®^; Tokyo, Japan) as a stain. The DFA-positive brain specimens were then transported on ice packs to OIE Rabies Reference Laboratory at Onderstepoort (South Africa), where DFA was repeated using FITC anti-rabies nucleocapsid conjugate (BIORAD^®^; CA, USA).

Out of the 278 brain tissue specimens, 23 (8.3%) were positive for rabies lyssaviruses.

### 2.5. RNA Extraction

One gram of each brain-infected tissue and a wild dog (*Lycaonpictus*) brain tissue (ReferenceLaboratory#811/97) previously shown to be RABV positive (and of the canid rabies biotype) were homogenized in sterile Eppendorf tubes. Total viral RNA was extracted using Tri Reagent (Sigma Aldrich, USA) according to the manufacturer’s instructions. The concentrations of the viral RNAs were measured using a NanoDrop spectrophotometer (NanoDrop Technologies; DE, USA) and stored at −70 °C until required for further analysis.

### 2.6. Detection of Nucleoprotein (N) Gene of Lyssaviruses Using Reverse Transcription Polymerase Chain Reaction (RT-PCR)

A combination of oligonucleotide primers, JW12_55–74_ (+) and 304_1514–1533_(-) and 550B_647–666_ (-) and Lys001_1–15_ (+), were used in this study, Table 1 [28,29] to target the full length or partial region of the N gene. The annealing positions and numbering of the oligos are based on that of Pasteur virus (PV) genome [30]. The primers were synthesized by Invitrogen Life Technologies (CA, USA), reconstituted with 10 mM Tris-HCl (pH 8.0) and 100 mM EDTA (TE) buffer (Promega; WI, USA), and stored frozen until utilized. The amplification of the N gene was performed using the GeneAmp^®^ 9700 (Applied Biosystems; MA, USA).

### 2.7. Reverse Transcription (RT)

Reverse transcription was done according to the protocol described by Markotter et al. [29]. Approximately 1 μg of the total viral RNA was heat-denatured at 65 °C for 5 min and annealed with 2 pmol/μL of JW12 and lys+, and then immediately snap cooled on ice. This was immediately followed by reverse transcription performed at 45 °C for 45 min in a 20 μL reaction containing 200 units of Murine Moloney Leukemia Virus Reverse Transcriptase (Invitrogen; CA, USA), 40 units of Riboblock^®^ribonuclease inhibitor (ThermoScientific; MA, USA), 10 mM of deoxynucleoside triphosphate (dNTP) mixture (Qiagen; Venlo, Netherlands), 0.1 M dithithreitol (DTT), and 5× M-MLV reaction buffer (Invitrogen; CA, USA). At the end of the reverse transcription reaction, the cDNA was inactivated at 85 °C for 5 min, snap cooled for 1 min, and then diluted two-fold with sterile nuclease free water and stored at −20 °C until required for use.

### 2.8. Polymerase Chain Reaction (PCR) Assay

The initial amplification of all the cDNA was performed using the primers JW12 (+) and 304 (-), targeting the full-length N gene [28]. An RNA sample extracted from a known rabies-positive sample of the canid variant of the RABV (an isolate from an African wild dog, *Lycaonpictus*, laboratory reference#: 811/97) was included as a positive control. Briefly, a 50 μL reaction mixture containing 2 μL of the cDNA, 1.25 units of Takara Taq DNA polymerase (Takara Biotechnology; Shiga, Japan), 1.5 mM MgCl_2_, 10 mM dNTP mixture, 40 pmol each of JW12 (+) and 304 (-), 1× Taq polymerase reaction buffer, and made up to 50 μL with nuclease free water. The amplification was carried out on an ABI 9700 thermocycler (Applied Biosystems; MA, USA), with an initial denaturation at 94 °C for 2 min, followed by 35 cycles of [94 °C for 30 s, 50 °C for 30 s, 72 °C for 120 s] and a final extension at 72 °C for 10 min. Those samples that did not amplify in the first round were subjected to hemi-nested PCR targeting a partial region of the N-terminal region of the N gene. This was achieved by using the same conditions as described above, except that a 1:500 dilution of first round PCR was used as the template and 40 pmoles of 550B (-) as the reverse primer. Avoidance of false-positive PCR results was done following standard precautionary measures [31].

### 2.9. Electrophoresis and Gel Image Documentation

The amplicons were visualized under UV transillumination after electrophoresis at 100 volts for 45 min through 1% ethidium bromide stained agarose gels (Labnet, Power Station 300, USA), with a 100 bp DNA ladder as the molecular weight marker (Promega, USA).

### 2.10. PCR Product Purification and Sequencing

The PCR products were purified using spin columns, (QIAquick^®^ PCR Purification kit, Netherlands) according to the manufacturer’s guidelines, and cycle sequenced bidirectionally. Analysis of the sequencing products was performed on an automated ABI 3100 DNA analyzer.

### 2.11. Phylogenetic Analysis

A consensus sequence of each RABV was obtained after the alignment of the forward and reverse sequences using algorithms in BioEdit [32]. Consensus sequences were trimmed to 1350 bp and 400 bp for the full and partial N regions, respectively. Multiple sequence alignment of the edited N region was performed using ClustalW [33]. Data analyses were conducted using Molecular Evolutionary Genetics Software Version 7 (MEGA7) [34]. The Kimura’s two parameter model was used to calculate the genetic distances between pairs of sequences [35]. The results were used to construct a maximum likelihood tree using MEGA7. Bootstrapping of a 1000 replicates was used to statistically evaluate the branching order of the phylogenetic tree. Bootstrap support of 70% was considered significant and provided evidence for phylogenetic grouping [36]. Partial N nucleotide sequences of lyssaviruses from this study, representative rabies viruses from other part of Nigeria, neighboring countries, Africa, and other continents (GenBank) were included in the analysis (Table 2). The phylogenetic tree was rooted using Lagos Bat lyssavirus and other distant rabies related lyssaviruses.

## 3. Results

### 3.1. Quality of the Extracted RNA

The purity ratio (quality) (260/280 nm) and concentration (ng/μL) of the extracted RNA showed that 19 RNA samples were between 1.6–1.8 nm and 1272.5–3538.5 ng/μL, respectively. However, three (14NG, 4NG, and 6NG) had purity ratios and nucleic acid concentration of 1.5 (710 ng/μL), 1.4 (432 ng/μL), and 1.5 (428 ng/μL). Only a single sample (2NG) had a purity ratio of 1.3 and nucleic acid concentration of 342 ng/μL.

### 3.2. Gel Electrophoresis Result

Twenty-two (96%) of the 23 specimens (brain specimens from slaughtered dogs, *n* = 21 and dogs presented to clinics as rabies-suspect, *n* = 2) tested yielded high-quality amplicons of the expected size of the N gene region analyzed. Of the 22, twelve specimens successfully amplified the full N and the rest (*n* = 10) amplified in the hemi-nested PCR.

### 3.3. Nucleoprotein Gene Features

The consensus sequences were deposited in GenBank, and were assigned accession numbers (Table 3).

All the nucleotide sequences obtained in this study belong to Africa 2 lineage. The topology of the phylogenetic trees for both nucleotide and amino acid sequences were similar (Figure 2 and Figure 3). A nucleotide sequence identity of 97.3% of the RABVs was obtained for this panel of viruses from dogs from Southeastern Nigeria and those from other parts of the country. Similarly, 97% amino acid sequence identity was observed between rabies viruses from Southeastern Nigeria and elsewhere in the country. The majority of the taxa (15 of the 19) clustered with RABV variants from the Plateau, Taraba, and Bauchi states. Of the 19 RABV nucleotide sequences obtained from this study, 15 were from Enugu State, and this is probably because of the existence of the largest dog market (OrieOrba) located in the State, and therefore more specimens were generally collected from there. A virus recovered from Enugu State (MN196576) clustered with lyssaviruses from the Niger Republic, and the node for this cluster was supported by a statistically significant bootstrap support value of 90%. Three viruses from dogs from Anambra State, MN196573, MN19573, and MN19574 formed a sub-cluster, albeit with a low bootstrap support value of 34%. However, this was no surprise, as these lyssaviruses originated from the same source location, thereby forming a sub-cluster (Figure 2).

## 4. Discussion

This study was undertaken with a view to detecting and genetically characterizing RABVs circulating among dogs in Southeastern Nigeria from the dog markets and other rabies-suspect dogs presented at the Veterinary Clinic. The RABV nucleotide sequences generated and deposited in the NCBI GenBank were dog RABVs from the three States of Southeastern Nigeria, namely; Anambra, Ebonyi, Enugu. Apart from Enugu State, where 4 RABV isolates have been sequenced and deposited in GenBank [25], this is the first comprehensive molecular analysis of RABVs from this region.

The phylogenetic analysis showed that the RABV nucleotide sequences obtained in this study all belong to Africa linage-2 [19,38] and are part of the greater rabies epidemiological cycle of the West African region. The data underscore the public health significance of rabies in this part of Africa, but within a unique set up of dog markets in Nigeria. The hazards should therefore be made known to the public, particularly the animal handlers or butchers, as there could be possible transmission from infected dogs to them. To date, there are no comprehensive reports of rabies in dog markets, and therefore this aspect of rabies epidemiology in rural Southeastern Nigeria will not be comprehensively known.

A detailed analysis of the phylogeographical structure of the Africa lineage-2 revealed strong population subdivisions at the country level, with only limited virus transmission among localities. RABV variants from Central and North Africa clustered into Africa 1a and 4 lineages, whereas those from the southern African countries were in Africa 1b and 3 clades. Clearly, all the RABV variants in this study were homogenous and closely related (99% sequence homology), suggesting the sharing of a common origin distinct to the outgroup. These data are consistent with the findings of Ogo et al. [12] who observed 99% nucleotide similarity amongst a panel of rabies viruses from Nigeria. This observation further supports the belief that rabies viruses from dogs in Nigeria belong to a single genetic lineage and a single major variant is maintained in this host species, albeit the low bootstrap values obtained. Ogunkoya et al. [39] demonstrated an identical pattern of reactivity amongst some Nigerian street rabies viruses recovered from domestic dogs from different geographical areas, suggesting antigenic homogeneity amongst the rabies viruses using monoclonal antibody (Mab) typing. This is also consistent with similar findings in other parts of the world, where identical reactivity patterns of rabies viruses were recovered from terrestrial species in contrast to the rabies viruses recovered from bats [40,41]. In southern Africa, a unique situation exists and different patterns of reactivity were observed between the canid and the mongoose rabies biotypes of southern Africa [42,43,44,45] also confirmed this observation with the antigenic analysis of rabies and rabies-related Mokola viruses from Zimbabwe. Nonetheless, molecular analyses provide a much finer delineation of pathogens, including rabies than antigenic typing.

The presence of smaller clades in the large cluster of viruses could indicate possible local outbreaks [42]. However, some of the variants from the same locality tend to group together, which could be attributed to a single and local rabies virus variant in that specific geographic zone. The site and geographic distribution patterns observed in this study could be attributed to the procurement and transportation of dogs from different parts of the country into dog markets in Southeastern Nigeria, thus promoting in-country rabies transmission.

The three viruses from IgboukwuAnambara state (MN196733, MN196574, and MN196575) formed a cluster and could be considered as a novel group, as this has not been observed previously. A RABV sequence (MN196576) obtained from a dog from Enugu State formed a cluster with nucleotide sequences from Niger Republic, with a bootstrap support of 90%. The phylogenetic evidence gathered in this study suggests that the RABV variant (MN196576) could have originated in Niger, and may have been introduced into Southeastern Nigeria through unrestricted movement of animals mediated by humans across the porous border and illegal importation of dogs involved in dog trade from the neighboring countries [11,46]. This further demonstrate show long-distance transmission of rabies is facilitated by human mediated animal movements [47,48]. This observation is not limited to Nigeria, as a RABV from Benin (U22485) clustered with one RABV from Cameroon (U22635); also Ghana RABVs (HM368162 and HM368096) clustered with Moroccan RABV (AY062090). Recently, a molecular study of rabies in Liberia demonstrated that rabies lyssaviruses from this country clustered with China lineage-2, indicating transcontinental transmission of rabies [20].

Three specimens (14NG, 4NG, and 6NG), although they tested positive with the DFA and also yielded amplicons of desired band size (606 bp), the resulting nucleotide sequence data was poor, underscoring the difficulty that routine rabies diagnostic laboratories may face when applying such advanced tests for rabies diagnosis. The one specimen (2NG) that was DFA positive but yielded no positive result following RT-PCR could be considered false positive, an occurrence that may result in the distortion of accurate burden of the disease on the continent. This is highly likely since specimens for rabies diagnosis reach the diagnostic laboratories in a state that may compromise testing due the specimen condition.

## 5. Conclusions

This study has provided an enhanced understanding of the molecular epidemiology of rabies in Southeastern Nigeria, and provided a baseline of the epidemiology and transmission dynamics of dog rabies in Nigeria. Some of rabies viruses from Nigeria that clustered with others from the Niger Republic underscore the transboundary transmission of rabies mediated by human movement between neighboring countries. These activities, whilst against the law, could also have serious public health concerns as zoonotic and transboundary diseases may be easily spread through such practices. The findings here probably suggest a multinational approach to control rabies as driven by Pan American Health Organization (PAHO) in South America, and only then is elimination of dog rabies on the continent towards 2030 a reality. Moreover, the results of this study have shown the presence of a novel rabies variant in Southeastern Nigeria, and also that all the rabies lyssaviruses sequenced in this study all belong to the Africa 2 lineage and are very closely related, indicating a common origin. Hence, national diagnostic laboratories should continually type positive rabies cases in order to understand the diversity of lyssavirus variants involved in rabies epizootiology in this region.

## Figures and Tables

**Figure 1 viruses-12-00134-f001:**
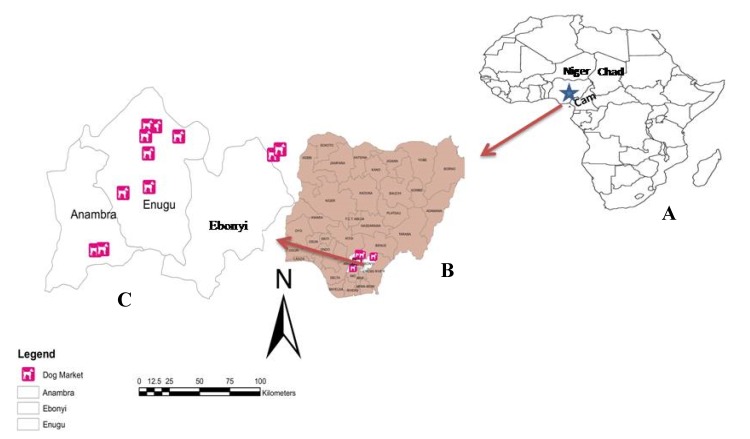
Map of West Africa showing Nigeria (**A**), South East region in the southeastern part of the country (**B**), and the five states of Southeastern Nigeria, showing areas of specimen collection in Anambra, Ebonyi, and Enugu States (**C**).

**Figure 2 viruses-12-00134-f002:**
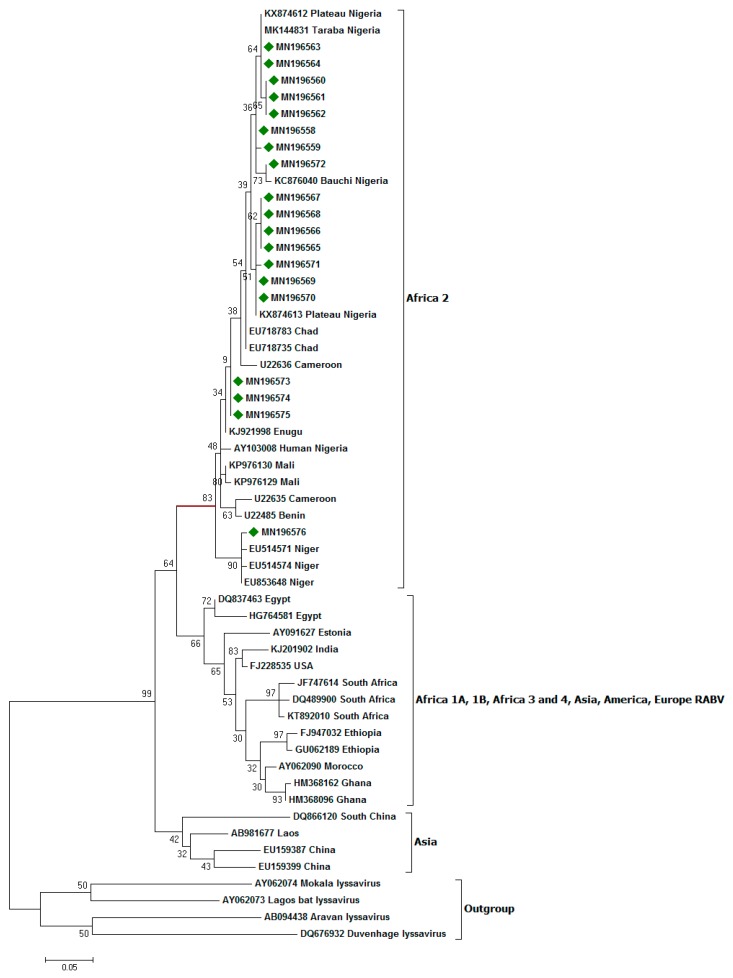
Maximum likelihood (ML) phylogenetic tree of the 19 Southeastern Nigerian RABV N-genes generated using ML algorithm (1000 bootstrap replications). The analyses involved 56 nucleotide sequences. All positions with <95% site coverage were eliminated. Evolutionary analyses were conducted in MEGA7 [34]. The bootstrap values (%) are shown on the nodes supporting these branches. The tree is drawn to scale, with branch lengths in the same units as those of the evolutionary distances used to infer the phylogenetic tree. The evolutionary distances were computed using the Kimura two-parameter method [37] and are in the units of the number of base substitutions per site. Isolates for which the partial N gene sequence was obtained in this study are indicated by a diamond green color.

**Figure 3 viruses-12-00134-f003:**
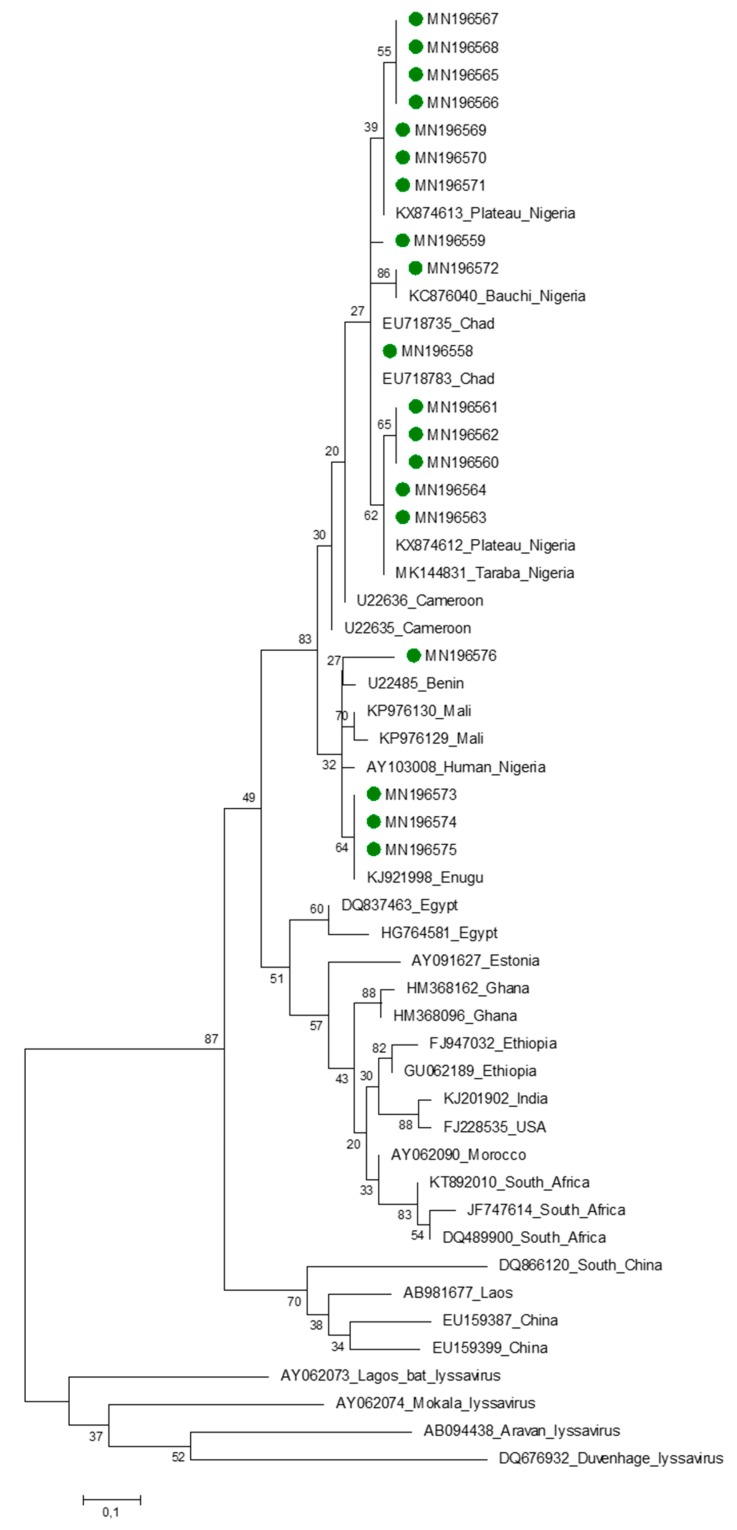
Maximum likelihood (ML) phylogenetic tree of the 19 Southeastern Nigerian RABV nucleoprotein deduced amino acid sequences generated using ML algorithm (1000 bootstrap replications).

**Table 1 viruses-12-00134-t001:** The Oligonucleotide primer sequences used in the study showing the annealing positions and their nucleotide sequences [28].

Oligonucleotide Sense	Nucleotide Sequence 5′-3′	Uses	Position on Genome
JW12 (+)	ATGTAACACC(C/T)CTACAATTG	cDNA synthesis, PCR, hnPCR	55–74
001lys (+)	ACGCTTAACGAMAAA	cDNA synthesis	1–15
304 (-)	TTGACAAAGATCTTGCTCAT	PCR and sequencing	1514–1533
550B (-)	GTRCTCCARTTAGCRCACAT	hnPCR	647–666

**Table 2 viruses-12-00134-t002:** Rabies diagnostic samples collected from Genbank for molecular and phylogenetic analysis.

Genbank Accession Number	Year of Collection	Country	Host Species	Lineage
MK144831	2014	Taraba, Nigeria	Dog	Africa 2
KX874612	2010	Jos, Nigeria	Dog	Africa 2
KX874613	2010	Jos, Nigeria	Dog	Africa 2
KC876040	2012	Bauchi, Nigeria	Cattle	Africa 2
KJ 921998	2012	Enugu, Nigeria	Dog	Africa 2
EU514571	2007	Niger	Dog	Africa 2
EU514574	2007	Niger	Dog	Africa 2
EU853648	1990	Niger	Dog	Africa 2
EU718783	2006	Chad	Dog	Africa 2
EU718735	2006	Chad	Dog	Africa 2
KP976130	2007	Mali	*Canis lupus familiaris*	Africa 2
KP976129	2007	Mali	*Canis lupus familiaris*	Africa 2
AY103008	1996	Nigeria	Human	Africa 2
U22636	1995	Cameroon	Dog	Africa 2
U22635	1995	Cameroon	Dog	Africa 2
U22485	1995	Benin	Dog	Africa 2
DQ837463	1999	Egypt	Dog	Africa 4
HG764581	2013	Egypt	Cattle	Africa 4
HM368162	2007	Ghana	Dog	Africa 1A
HM368096	2008	Ghana	Dog	Africa 1A
AY062090	2001	Morocco	Fox	Africa 1B
FJ947032	2009	Ethiopia	Dog	Africa 1A
GU062189	2009	Ethiopia	*Canissimensis*	Africa 1A
AY091627	2002	Estonia	Raccoon dog	Europe
KJ201902	2010	India	*Musmusculus*	Asia
FJ228535	1950	USA	Dog	America
JF747614	2008	South Africa	Canine	Africa 3
KT892010	1997	South Africa	African wild dog	Africa3
DQ194892	1997	South Africa	African wild dog	Africa 3
DQ 866120	2006	South China	Dog	Asia (China)
AB981677	2012	Laos	*Canis lupus familiaris*	Asia
EU159399	1994	China	Dog	Asia
AB094438	2002	Kyrgyzstan	*Myotisblythii*	Aravanlyssairus (Outgroup)
DQ676932	2006	South Africa	Human	Duvenhagelyssavirus (Outgroup)
AY062073	2002	South Africa	Bat	Lagos bat Lyssavirus (Outgroup)
AY062074	2001	Nigeria	Shrew	Mokolalyssavirus (Outgroup)

**Table 3 viruses-12-00134-t003:** Rabies specimens collected from brain tissues of dogs (*Canisfamiliaris*) in Southeastern Nigeria for molecular and phylogenetic analysis.

S/N	Lab ID	Date of Collection	Source Location	Accession Number
1	14NG	11 October 2015	OrieOrba, Enugu State, Nigeria	MN196558
2	11NG	26 October 2015	OrieOrba, Enugu State, Nigeria	MN196559
3	18NG	7 November 2015	OrieOrba, Enugu State, Nigeria	MN196560
4	12NG	7 November 2015	OrieOrba, Enugu State Nigeria	MN196561
5	7NG	6 December 2015	OrieOrba, Enugu State Nigeria	MN196562
6	17NG	10 December 2015	OrieOrba, Enugu State Nigeria	MN196563
7	1NG	19 December 2015	OrieOrba, Enugu State Nigeria	MN196564
8	20NG	9 January 2016	NkwoOgbede Enugu State, Nigeria	MN196565
9	26NG	10 January 206	IheNze, Enugu State, Nigeria	MN196566
10	28NG	17 January 2016	IheNze, Enugu State, Nigeria	MN196567
11	29NG	22 February 2016	Umuna, Enugu State, Nigeria	MN196568
12	23NG	18 April 2016	Iboko, Ebonyi State, Nigeria	MN196569
13	13NG	28 April 2016	Nkwoogbede, Enugu State	MN196570
14	19NG	16 May 2016	OrieOrba, Enugu State Nigeria	MN196571
15	22NG	16 May 2016	OrieOrba, Enugu State Nigeria	MN196572
16	21NG	29 May 2016	NkwoIgboukwuAnambra State, Nigeria	MN196573
17	27NG	7 June 2016	NkwoIgboukwuAnambra State, Nigeria	MN196574
18	25NG	7 June 2016	NkwoIgboukwuAnambra State, Nigeria	MN196575
19	16NG	20 July 2016	Vet Clinic Uwani Enugu, Nigeria	MN196576

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
