# Peer review of "Molecular Detection of Rabies Lyssaviruses from Dogs in Southeastern Nigeria: Evidence of TransboundaryTransmission of Rabies in West Africa"

_viruses, 2020, doi:10.3390/v12020134_

Round 1
Reviewer 1 Report
The manuscript by Eze and colleagues describes molecular characteristics of rabies viruses isolated from dogs in southeastern Nigeria. The manuscript is generally well written and provides valuable insights into the molecular epidemiology in West Africa. The fact that the viruses derived from dog markets and resturants raises a point that I strongly advise the authors to comment upon in the Discussion or Conclusions sections: This situation must present a significant risk of transmission to humans during butchering or even consumption – are there known cases of human rabies associated with infected dogs in the markets?
The manuscript would benefit considerably from revision to address a number of minor, but it think important, issues:
The term South Eastern (e.g. in the title and elsewhere) is much more commonly presented as southeastern (one word).
Line 34: Case fatality rate APPROACHING OR ESSENTIALLY 100% (there have been a handful of survivors).
Line 36: inapproximately 10,000 annual human deaths: in [space] approximately. This figure is likely too high for deaths just in Nigeria. Reference (1) is from 1978 and reference 2 only has authors listed so it cannot be checked for validity. I did not check all the references, but based on Reference 2 and 18, the authors clearly should check reference format and that the citations in the text are for the correct reference.
Line 43: Nigeria is one huge country -> Nigeria is a large country
Line 46: I don’t understand how “smuggling, human, drug and illicit arms trafficking” promote transmission of rabies virus. Similarly, on lines 271-272, there is an unwarranted implication that rabies viruses may have been transmitted by humans between Niger and Nigeria, when transmission via dogs is much more likely, as indicated immediately following.
Reference 18 is supposed to support the 3 African lineages, but instead refers to Artic and Greenland rabies viruses
Lines 112-114: Poorly constructed sentence that should be re-written.
Line 112 and Table 1: Are domestic dogs not Canis lupus familiaris?? What do you mean by “wild dog” – is this the African wild dog Lycaon pictus?
Figures 2 and 3: Nice gels but it seems that showing gels of PCR products in manuscripts has not been common for a number of years and the figures do not add significantly to the manuscript. Consider deleting these figures.
Author Response
The manuscript by Eze and colleagues describes molecular characteristics of rabies viruses isolated from dogs in southeastern Nigeria. The manuscript is generally well written and provides valuable insights into the molecular epidemiology in West Africa. The fact that the viruses derived from dog markets and resturants raises a point that I strongly advise the authors to comment upon in the Discussion or Conclusions sections: This situation must present a significant risk of transmission to humans during butchering or even consumption – are there known cases of human rabies associated with infected dogs in the markets?
Response: This has been addressed in line 218-219. Yes, there are speculations of human rabies associated with infected dogs in the market; however, none of these cases was confirmed. Also, this research focused on dog rabies in dog market specifically.
The term South Eastern (e.g. in the title and elsewhere) is much more commonly presented as southeastern (one word).
Response: It has been corrected throughout the manuscript. Thank you
Line 34: Case fatality rate APPROACHING OR ESSENTIALLY 100% (there have been a handful of survivors)
Response: it has been corrected to approaching 100%. Thank you
Line 36: inapproximately 10,000 annual human deaths: in [space] approximately. This figure is likely too high for deaths just in Nigeria. Reference (1) is from 1978 and reference 2 only has authors listed so it cannot be checked for validity. I did not check all the references, but based on Reference 2 and 18, the authors clearly should check reference format and that the citations in the text are for the correct reference.
Response: The annual deaths have been corrected from 10,000 to 1637. The second reference has been corrected. The problem was that reference 2 was mistakenly divided into two which formed reference 3 thus distorting the bibliography and then making the citation in the text incorrect. But this has been taken care of.
Line 43: Nigeria is one huge country -> Nigeria is a large country
Response: This has been corrected to a large country.
Line 46: I don’t understand how “smuggling, human, drug and illicit arms trafficking” promote transmission of rabies virus. Similarly, on lines 271-272, there is an unwarranted implication that rabies viruses may have been transmitted by humans between Niger and Nigeria, when transmission via dogs is much more likely, as indicated immediately following.
Responses: These have been corrected
Reference 18 is supposed to support the 3 African lineages, but instead refers to Artic and Greenland rabies viruses
Responses: The correction of the references has taken care of this.
Lines 112-114: Poorly constructed sentence that should be re-written.
Response: The sentence has been re-written
Line 112 and Table 1: Are domestic dogs not Canis lupus familiaris?? What do you mean by “wild dog” – is this the African wild dog Lycaon pictus?
Response: It has been corrected to Lycaon pictus
Figures 2 and 3: Nice gels but it seems that showing gels of PCR products in manuscripts has not been common for a number of years and the figures do not add significantly to the manuscript. Consider deleting these figures.
Response: Figures 2 and 3 have been deleted
Reviewer 2 Report
No comments
Author Response
We would like to thank you for facilitating the review of our manuscript on” Molecular detection of rabies lyssaviruses from dogs in Southeastern Nigeria: evidence of transboundary transmission of rabies in West Africa”.
Reviewer 3 Report
This manuscript by UU Eze et al on the molecular detection of rabies viruses from dogs in South eastern Nigeria is an interesting, concise and fairly well-written manuscript.
I have a few minor comments/ suggestions for the authors.
Line 68: “The study was designed to diagnose RABVs in dogs…”
A better term to use would be “to diagnose rabies infection in dogs” or “detect RABV” (Since we generally diagnose infections/diseases and detect viruses/pathogens)
In addition to the phylogenetic analysis of the nucleotide sequences, the deduced amino acid sequences of the N gene of these RABVs could have been compared. This would have provided valuable insights, without any additional experiments. There are several errors in numbering of the references in the text: To point out just a few,
Line 105: Ref should be 28 (instead of 27); Line 154: Ref should be 32 (instead of 31); Line 121: Ref should be 29,30
There are some minor grammatical/punctuation/typographical errors through-out the manuscript, which should be rectified. To name a few- Lines 187-188, Lines 293-294
Author Response
This manuscript by UU Eze et al on the molecular detection of rabies viruses from dogs in South eastern Nigeria is an interesting, concise and fairly well-written manuscript.
I have a few minor comments/ suggestions for the authors.
Line 68: “The study was designed to diagnose RABVs in dogs…”A better term to use would be “to diagnose rabies infection in dogs” or “detect RABV” (Since we generally diagnose infections/diseases and detect viruses/pathogens)
Response: the term has been changed to detect RABV.
In addition to the phylogenetic analysis of the nucleotide sequences, the deduced amino acid sequences of the N gene of these RABVs could have been compared. This would have provided valuable insights, without any additional experiments.
Response: this has been included. The amino acid sequences topology is the same as that of the nucleotide sequences
There are several errors in numbering of the references in the text: To point out just a few,Line 105: Ref should be 28 (instead of 27); Line 154: Ref should be 32 (instead of 31); Line 121: Ref should be 29,30
Response: The problem was that reference 2 was mistakenly divided into two which formed reference 3 thus distorting the bibliography and then making the citation in the text incorrect. But this has been taken care of.
There are some minor grammatical/punctuation/typographical errors through-out the manuscript, which should be rectified. To name a few- Lines 187-188, Lines 293-294 .
Response: This has been corrected. Also a general check of typographical and grammatical error has been made,improving the manuscript.